# Evaluation of Fluoride Adsorption Mechanism and Capacity of Different Types of Bone Char

**DOI:** 10.3390/ijerph18136878

**Published:** 2021-06-26

**Authors:** Benyapa Sawangjang, Phacharapol Induvesa, Aunnop Wongrueng, Chayakorn Pumas, Suraphong Wattanachira, Pharkphum Rakruam, Patiparn Punyapalakul, Satoshi Takizawa, Eakalak Khan

**Affiliations:** 1International Postgraduate Program in Environmental and Hazardous Waste Management, Graduate School, Chulalongkorn University, Bangkok 10300, Thailand; benyapa@env.t.u-tokyo.ac.jp; 2Faculty of Environment and Resource Studies, Mahidol University, Nakhon Pathom 73170, Thailand; induvesa.p@hotmail.com; 3Department of Environmental Engineering, Faculty of Engineering, Chiang Mai University, Chiang Mai 50200, Thailand; suraphong@eng.cmu.ac.th (S.W.); pharkphum@eng.cmu.ac.th (P.R.); 4Research Program in Control of Hazardous Contaminants in Raw Water Resources for Water Scarcity Resilience, Center of Excellence on Hazardous Substance Management (HSM), Bangkok 10300, Thailand; 5Center of Excellence in Bioresources for Agriculture, Industry and Medicine, Department of Biology, Faculty of Science, Chiang Mai University, Chiang Mai 50200, Thailand; chayakorn.pumas@gmail.com; 6Department of Environmental Engineering, Faculty of Engineering, Chulalongkorn University, Bangkok 10300, Thailand; patiparn387@gmail.com; 7Department of Urban Engineering, Graduate School of Engineering, The University of Tokyo, Tokyo 113-8654, Japan; takizawa@env.t.u-tokyo.ac.jp; 8Department of Civil and Environmental Engineering and Construction, University of Nevada, Las Vegas, NV 89154-4015, USA; eakalak.khan@unlv.edu

**Keywords:** bone char, fluorosis, hydroxyapatite

## Abstract

The fluoride adsorption capacity of three types of bone char (BC), including cow BC (CBC), chicken BC (CKBC), and pig BC (PBC), was examined. At the optimum charring conditions (temperature and time), PBC had the highest hydroxyapatite (HAP) content (0.928 g-HAP/g-BC), while CBC had the highest specific surface area (103.11 m^2^/g-BC). CBC also had the maximum fluoride adsorption capacity (0.788 mg-F/g-HAP), suggesting that fluoride adsorption capacity depends more on the specific surface area of the BC than the HAP content. The adsorption data of CBC, CKBC, and PBC fit well with the pseudo-second-order model and the Langmuir isotherm. The maximum fluoride adsorption capacity of BC reached the maximum value when the solution had a pH of approximately 6.0. Lastly, the highest fluoride desorption occurred when the BCs were soaked in solutions with a pH higher than 11.0.

## 1. Introduction

A substantial percentage of the population around the world, especially in South and Southeast Asia, perceive groundwater to be of better quality than surface water (in terms of microbial contamination), and thus prefer it as a source of drinking water [1,2,3]. However, groundwater may be contaminated by excessive concentrations of arsenic and fluoride [4,5]. Fluoride contamination in groundwater originates mainly from the dissolution of natural minerals in rocks [6]. The effects of fluoride on human health depend on its concentration in the drinking water ingested by individuals. Fluoride is effective in preventing tooth decay, especially among children, when the concentrations are less than 0.5 mg/L [7,8]. However, higher fluoride concentrations are known worldwide to cause serious health problems, such as dental and skeletal fluorosis. Moreover, long-term fluoride accumulation in humans can lead to cancer [9,10,11]. Therefore, the World Health Organization (WHO) set the threshold fluoride concentration in drinking water at 1.5 mg/L [12].

Several methods have been applied for the defluoridation of groundwater, such as co-precipitation, adsorption, ion exchange, and reverse osmosis [13,14,15]. Among them, adsorption is a readily applicable and economical method [16,17]. Both natural and synthetic adsorbents, such as activated clay, ion exchange resin, activated alumina, and different types of bone char (BC), have been applied for fluoride removal in developing countries [18,19,20]. BC is a natural adsorbent made from bone waste and can adsorb fluoride ions in water mainly because of its hydroxyapatite (HAP) (Ca_10_(PO_4_)_6_(OH)_2_) content [21,22]. BC removes fluoride in groundwater through the exchange between hydroxyl and fluoride ions according to the following chemical reaction [23,24].
(1)$${\mathrm{Ca}}_{10}{\left({\mathrm{PO}}_{4}\right)}_{6}{\left(\mathrm{OH}\right)}_{2}+{2F}^{-}\to \;{\mathrm{Ca}}_{10}{{(\mathrm{PO}}_{4})}_{6}F_{2}+{2\mathrm{OH}}^{-}$$

The application of BC for defluoridation could result in an unpleasant taste, smell, and color (yellowish) in the treated water, which are caused by the remaining organic matter in the bones when the charring temperatures are below 500 °C [25]. Leyva et al. observed that at temperatures higher than 700 °C, de-hydroxylation of HAP in BC occurred, which reduced its fluoride adsorption capacity [26]. Furthermore, the conditions of the charring process (temperature and time for charring) influenced the fluoride adsorption capacity due to the differences in crystallinity and surface properties [27].

In this study, we investigated the optimum charring-process conditions among different types of BCs, and the roles of the specific surface area and HAP content on the fluoride adsorption capacity. The three types of BCs, including cow, chicken, and pig BCs, were studied. The charring temperatures of 450, 550, and 650 °C, and charring times of 1, 2, and 3 h at each temperature were examined. Then, the HAP content and the specific surface area of those BCs were measured.

## 2. Materials and Methods

### 2.1. BC Synthesis

To produce the three types of BCs, raw bones of cow, chicken, and pig were purchased from a fresh market in Chiang Mai province, Thailand, and were processed as follows. The fats remaining inside the bones were taken out, and the bones were rinsed with deionized water. Then, the cleaned bones were dried in an oven at 110 °C for 24 h. After drying, the bones were crushed by a machine into small pieces (around 2–5 cm). The crushed bones were placed in ceramic cups with lids to limit the amount of oxygen during the charring process. The cups were then placed in a furnace and charred under nine different conditions (charring temperatures of 450, 550, and 650 °C, and charring times of 1, 2, and 3 h at each temperature). The chosen charring temperatures and charring times were conducted based on on-site carbonization and actual application; however, the given temperature of higher than 650 °C and the charring time of more than 3 h could not produce a good quality of BC. This was due to a leakage of oxygen into the furnace during the charring process. After charring, the BC samples were further crushed by a hammer into pieces small enough to pass through a No. 60–40 mesh sieve (250–420 µm).

### 2.2. Characterization of BC

The HAP content of each BC was analyzed by X-ray Diffraction (XRD, BRUKER model D8 Advance, Germany). Meanwhile, the surface areas were estimated using the nitrogen adsorption method and the Brunauer–Emmett–Teller (BET) equation (Autosorb 1 MP, Quantachrome Instrument, Boynton Beach, FL, USA). The points of zero charge (PZC) were determined as follows [28]. The BCs were washed with deionized water and then dried in an oven at 110 °C for 12 h. For each type of BC, 200 mg was soaked in 100 mL of deionized water, with the pH adjusted between 3.0 and 12.0 using nitric acid or sodium hydroxide. Then, the mixtures were shaken at 120 rpm for 24 h using a shaker (GFL, Orbital shaker 3017, Germany). After shaking, the final pH of each sample was measured. The initial and final pH values were plotted on a scatter diagram. The PZC was obtained at the crossing point between the lines connecting the pH data and the diagonals connecting the equal initial and final pH (pH_PZC_).

### 2.3. Adsorption Kinetics and Isotherm of BC

Synthetic groundwater with approximately 20 mg/L fluoride concentration was used in the batch adsorption experiments. This value is equal to the fluoride concentration of groundwater in an actual contaminated site at Banbuakkhang School in Chiang Mai Province, Thailand. The synthetic groundwater was prepared by dissolving sodium fluoride (NaF) in deionized water. The adsorption kinetics experiments were conducted by mixing 20 g/L BC into 250 mL synthetic groundwater with a pH of 7.0 (controlled by a phosphate buffer). These mixtures were shaken using the rotary shaker at 200 rpm and 24–36 °C. The shaking was stopped at different adsorption times between 0 and 24 h, and the BCs were separated from the solution using a nylon syringe filter (dia. 13 mm, nominal pore size 0.45 µm, Chrom Tech, Apple Valley, MN, USA). The filtrates were analyzed for residual fluoride concentration. All water samples were measured for fluoride concentration by using the colorimetric SPADNS method (Standard Methods: 4500-F-D) at a wavelength of 570 nm (Jenway 6400 Spectrophotometer, Jenway, UK).

The adsorption isotherms were studied by varying the initial fluoride concentrations from 0 to 16 mg/L (in accordance with fluoride concentration in groundwater in the study area), which were then added to 20 g/L BC with a pH of 7.0 (controlled by a phosphate buffer). The equilibrium time was based on the kinetics study results.

### 2.4. Fluoride Adsorption at Different pH

Adsorption experiments were conducted for solutions with pH from 4.0 to 12.0 varied by 0.1 N nitric acid or 0.1 N sodium hydroxide. The initial fluoride concentration was 20 mg/L. Then, 2 g of BC was shaken in 100 mL of water at 200 rpm until equilibrium was reached. After that, the samples were filtered through a nylon membrane filter (the same filter mentioned in the preceding subsection). The solutions were analyzed for the remaining fluoride concentration.

### 2.5. Desorption of Fluoride Ions

After reaching the adsorption equilibria, the BCs were separated from the mixtures by filtration through nylon membrane filters (the same filter used in Section 2.3). The separated BCs that contain fluoride were soaked in 100 mL of water with the pH adjusted between 6.0 and 11.0 using 0.1 N of nitric acid or 0.1 N of sodium hydroxide. Then, the desorbed fluoride concentration was analyzed by measuring the fluoride concentration in the aqueous phase.

### 2.6. Adsorption Kinetics and Isotherms

Different kinetic models, including pseudo-first-order and pseudo-second-order equations [29,30], were applied to quantitatively establish the rate of defluoridation by BC.

#### 2.6.1. The Pseudo-First-Order Equation

The pseudo-first-order equation is given by:(2)$$\;\ln\left(\frac{q_{e}}{q_{e}-q_{t}}\right)=k_{p1}t$$
which can be rearranged in linear form as follows:(3)$$\log\left(q_{e}-q_{t}\right)=\log(q_{e})-\frac{k_{p1}t}{2.303}$$
where $q_{t}$ and $q_{e}$ is the adsorption capacity at any given time t and at equilibrium (mg/g), respectively, t is the time (min), and *K_p1_* is the pseudo-first-order rate constant (min^−1^).

#### 2.6.2. Pseudo-Second-Order Equation

The pseudo-second-order equation is generally applied to describe the chemical reactions of heterogeneous materials. This model is given by:(4)$$\;\frac{t}{\;q_{t}}=\frac{1}{k_{p2}q_{e}^{2}}+\frac{t}{q_{e}}$$
where $q_{t}$, $q_{e}$, and t are defined similar to those in Equation (3), and $k_{p2}\;$ is the pseudo-second-order rate constant (g/mg∙min).

#### 2.6.3. Adsorption Isotherms

The adsorption isotherms were characterized by comparing data obtained with the Langmuir and Freundlich models. The Langmuir isotherm describes monolayer sorption onto the surface, with the sorption occurring only on some sites. There are no interactions between the molecules. The model equation is given by [31], which can be rearranged in a linear form as:(5)$$\frac{1}{q_{e}}=\frac{1}{K_{L}q_{0}}\frac{1}{C_{e}}+\frac{1}{q_{0}}$$
where $q_{0}$ is the amount of fluoride adsorbed per unit weight of BC that forms a complete monolayer on the surface (mg/g), $q_{e}$ is the total amount of fluoride adsorbed per unit weight of BC at equilibrium (mg/g), $C_{e}$ is the fluoride concentration in the solution at equilibrium (mg/L), and $K_{L}$ is a constant related to the energy of sorption (L/mg).

The Freundlich isotherm can be applied to non-ideal sorption on heterogeneous surfaces and multilayers. The model is given by [31], which can be rearranged in a linear form as:(6)$$\;\log q_{e}=\frac{1}{n}\log C_{e}+\log K_{F}$$
where $q_{e}$ is the total amount of fluoride adsorbed per unit weight of BC at equilibrium (mg/g), $C_{e}$ is the fluoride concentration in the solution at equilibrium (mg/L), and $K_{F}$ and n are the Freundlich constants related to the adsorption capacity (L/g) and the adsorption concentration (dimensionless), respectively.

## 3. Results

### 3.1. BC Adsorbent Characteristics

#### 3.1.1. HAP Content of BCs

The HAP contents in the BCs were determined by XRD. The results are shown in Figure 1. The optimum charring conditions were 650 °C charring temperature and 3 h charring duration to produce the highest HAP content of PBC at 0.93 g-HAP/g-BC and CKBC at 0.85 g-HAP/g-BC. CBC had the highest HAP content (0.63 g-HAP/g-BC) when charred at 550 °C for 3 h.

#### 3.1.2. Textural Properties of BCs

The textural properties for PBC, CKBC, and CBC charred at the optimal conditions are summarized in Table 1. The CBC adsorbent had the highest specific surface area (103.11 m^2^/g). Meanwhile, the CKBC adsorbent had the lowest specific surface area among the three BC adsorbents.

#### 3.1.3. Points of Zero Charge of BCs

The PZC values of the PBC, CKBC, and CBC adsorbents were 8.6, 9.0, and 7.9, respectively, as shown in Figure 2. The surface of an adsorbent is positively charged when the solution pH is less than the PZC, whereas the surface becomes negatively charged when the solution pH is above the PZC.

### 3.2. Effect of Solution pH

Figure 3a shows the relationship between the initial pH of the solution and the final pH after the adsorption process. The final pH decreased when the initial pH was higher than the pH_PZC_. Figure 3b shows that fluoride adsorption is low when the initial pH is near or higher than pH_pzc_ due to proton release (deprotonation).

### 3.3. Kinetic Adsorption of BC Adsorbents

The pseudo-first-order and the pseudo-second-order models were applied to identify the kinetics of fluoride adsorption on the BCs. Figure 4 illustrates the kinetic adsorption of fluoride on the PBC, CKBC, and CBC adsorbents at a solution pH of 7.0. All BCs adsorbed fluoride rapidly in the first 10 min, and then gradually slowed down until equilibrium was reached in less than 1 h. At equilibrium, the highest fluoride adsorption capacities of PBC, CKBC, and CBC adsorbents were 0.366, 0.347, and 0.497 mg F/g-BC, respectively. The results show that CBC adsorbs more fluoride than PBC and CKBC, even if it had the lowest HAP content among the three. The calculated surface areas of HAP for each BC are summarized in Table 2.

To investigate the rate constant of fluoride adsorption on the PBC, CKBC, and CBC adsorbents, the data obtained from the experiments were fitted with two kinetic models. The correlation coefficients (*R*^2^) for the linearized pseudo-second-order model fit were higher than that of the pseudo-first-order model. The calculated parameters for each model are summarized in Table 3. The theoretical fluoride adsorption capacities derived from the pseudo-second-order kinetic model were similar to the experimental values for PBC, CKBC, and CBC.

### 3.4. Adsorption Isotherm of BC Adsorbents

As demonstrated in Figure 5, the fluoride adsorption capacity significantly increased with fluoride concentration initially. PBC and CBC reduced the fluoride concentration from 5 mg/L to below the WHO threshold level (1.5 mg/L) by using 20 g/L BC. Meanwhile, CKBC could not reduce the fluoride concentration to below the WHO threshold. We applied the Langmuir and the Freundlich isotherm models to describe the results of the adsorption experiment (Figure 5). The adsorption isotherm parameters evaluated using the Langmuir and the Freundlich models are summarized in Table 4.

### 3.5. Desorption of Fluoride Adsorbed on BC Adsorbents

Figure 6 illustrates the relationship between the percentages of fluorides desorbed from the BC adsorbents and the pH solution used for the desorption process (pH 6.0 to 11.0). The highest fluoride desorption occurred when the BC was soaked in a solution with a pH higher than 11.0.

## 4. Discussion

From the results of the HAP contents in the BCs, it can be inferred that the HAP in the CBC was destroyed when the charring temperatures were higher than 550 °C. The fluoride adsorption capacity reduced because of the dehydroxylation of the hydroxyapatite structure at high temperatures for charring conditions [26]. A high HAP content can promote fluoride adsorption [21]. However, the analysis of variance (ANOVA) results indicate that the mean HAP contents were significantly different among the three types of BCs (*p* < 0.05). The HAP content of CKBC was significantly higher than those in PBC and CBC (*t*-test, *p* < 0.05). The duration of charring time and the temperatures significantly affected the HAP content (ANOVA, *p* < 0.05). Moreover, the weight loss quantities during the charring process were significantly different in the three types of bones (ANOVA, *p* < 0.05). The best charring conditions for synthesizing PBC and CKBC was 650 °C for 3 h, and 550 °C for 3 h for CBC. The CKBC produced significantly higher HAP than other BCs in this research.

An analysis of the BC textural properties suggest that specific surface area and pore volume are related to the fluoride adsorption capacity [32]. Particularly, an adsorbent with a high specific area tends to have high fluoride adsorption capacity. Leyva et al. reported that higher specific surface area and total pore volume of BC result in higher fluoride adsorption efficiency [26]. In this study, CKBC produced significantly higher HAP than other BCs in this work. In contrast, CKBC had the lowest specific surface area when compared to the others. Thus, CKBC has the lowest fluoride adsorption capacities than other types of BC.

In this study, PZC values of PBC, CKBC, and CBC adsorbents were 8.6, 9.0, and 7.9, respectively. Medellin–Castillo et al. reported a PZC value of BC at pH 8.4 [33]. At a pH equal to the PZC, the surface charges of an adsorbent are neutral or nearly zero, which implies that the surface charge of BC depends mainly on the interaction between the BC surface and the ions in the solution. Here, PBC, CKBC, and CBC were used to adsorb the fluoride ions. Therefore, a positively charged surface on the adsorbents is favorable in attracting anions. The pH of the water sample in this study was in the range of 6.5 to 7.0. Chuah et al. [34] reported that the pH values of groundwater from deep and shallow wells in Chiang Mai Province were 7.04 ± 0.78 and 7.11 ± 0.69, respectively. These pH values are less than PZC values of the three types of BCs investigated here. Thus, fluoride removal by BCs should be more favorable from acidic groundwater. Moreover, the specific fluoride adsorption capacity, in terms of fluoride adsorbed (mg) per HAP content (g), tends to increase with the HAP surface area (as shown in Table 2). Therefore, it can be concluded that both the HAP contents and the surface area of the adsorbents affected the fluoride adsorption capacity.

The fluoride adsorption capacity on all BCs rapidly increased in the first 10 min, then reached the equilibrium stage within 1 h. CBC had the highest fluoride adsorption capacities compared to the others due to it containing the highest HAP content. At equilibrium during the kinetic adsorption tests, the pseudo-second-order kinetic model represented the fluoride adsorption on all BC adsorbents well, rather than the pseudo-first-order model. Based on the theoretical fluoride adsorption capacities, the pseudo-second-order kinetic model was related to the experimental values that were directly obtained from the laboratory experiment in this study. It can be indicated that the adsorption of fluoride on the BC surface best fit the pseudo-second-order kinetic model. The pseudo-second-order kinetic model indicated chemisorption of fluoride ions on the BC surface; that is, valence forces and ion exchange between the fluorides and the PBC, CKBC, and CBC adsorbents could have occurred [31]. Additionally, the fluoride adsorption data on the PBC, CKBC, and CBC adsorbents fit better with the Langmuir isotherm model than the Freundlich isotherm model based on the *R*^2^ values. This indicates that the adsorbed fluoride ions on the PBC, CKBC, and CBC adsorbents formed monolayers [31,32]. Previous studies have reported that the Langmuir sorption isotherm can be used to describe the adsorption process of fluoride ions on BCs [25,35]. Furthermore, Alkurdi and colleagues (2020) also reported that the fluoride removal on bone char followed the pseudo-second-order kinetic model, and that the Langmuir model is the best model to represent fluoride removal on bone char [36]. However, it was also observed that the pseudo-first-order kinetic model was best fitted for arsenic adsorption onto bone char at a low initial concentration of around 0.5 mg/L, whereas the arsenic removal followed the pseudo-second-order model at high arsenic concentrations (2.5–5 mg/L). Thus, this indicates that the rate of adsorption on inorganic substances, i.e., arsenic removal on bone char, was significantly affected by the initial concentration of inorganic substances [37].

The solution pH significantly affected the fluoride adsorption capacity of the BCs. Particularly, the fluoride adsorption capacity decreased substantially with increasing pH from 3.0 to 11.0 [35]. The results of this study showed a similar trend, as shown in Figure 5. This result suggests that protons were released when pH > pH_PZC_, as detailed in Equations (7) and (8) [33]. Thus, there is an insufficient amount of hydroxyl ions to be replaced by fluoride ions. In contrast, when pH < pH_PZC_, the final pH increased through the adsorption of protons. Therefore, sufficient hydroxyl ions were available for replacement by fluoride ions. However, when the initial pH value was further reduced to around 4.0, the proton concentration increased per Equations (9) and (10). This reaction interferes with the ligand exchange between the fluoride and hydroxyl ions, which is why the fluoride adsorption capacity varies at different initial pH values. The fluoride adsorption capacity increased to the maximum value when the initial pH was approximately 6.0, owing to the suppressed deprotonation and extremely low competition between fluoride and hydroxyl ions as a ligand to the P surface. However, when the pH was further reduced to approximately 4.0, the protonation of the P surface prevented the OH ligand exchange with fluoride. As a result, the fluoride adsorption decreased when the initial pH was 4.0.

The deprotonation reaction is governed by:(7)$$\equiv P-\mathrm{OH}\to \;\equiv {\mathrm{PO}}^{-}+H^{+}$$
(8)$$\equiv \mathrm{Ca}-\mathrm{OH}\to \;\equiv \mathrm{Ca}-O^{-}+H^{+}$$
while the protonation reaction is governed by:(9)$$\equiv P-\mathrm{OH}+{\;H}^{+}\to \;\equiv {\mathrm{POH}}_{2}^{+}$$
(10)$$\equiv \mathrm{Ca}-\mathrm{OH}+{\;H}^{+}\to \;\equiv \mathrm{Ca}-{\mathrm{OH}}_{2}^{+}$$
where “≡” represents the HAP surface.

The highest fluoride desorption was observed when the BCs were soaked in solutions with a pH higher than 11.0. However, when the pH was near or less than PZC, the fluoride desorption was either minimal or unnoticeable in the water. Medellin–Castillo et al. observed similar results [35] when they evaluated fluoride desorption of CBC that was loaded into an aqueous solution with pH from 7.0 to 12.0. The desorption results could refer to a way to regenerate fluoride adsorption capacity of the adsorbent. It is necessary to examine the reusability of the adsorbent in order to reduce the generation cost and waste generation in the future.

In addition, several studies have reported that hydroxyapatite is an alternative material for fluoride removal. Hydroxyapatite can remove fluoride by exchanging between OH-group on hydroxyapatite and fluoride ion in water [14,21,33,38].

Sani and colleagues (2016) compared the fluoride adsorption capacity on nano-hydroxyapatite/stilbite (nHAST) and bone char (BC), which is the adsorbent that contained the hydroxyapatite content. They prepared the nHAST based on locally available stibnite zeolites, which have a large particle size. The results of the fluoride adsorption capacity, which normalized into the amount of hydroxyapatite (HAP) on the adsorbent, illustrated that the nHAST (9.15 mg F^−^/g HAP) was significantly higher than fluoride adsorbed on BC (1.08 mg F^−^/g HAP). It was a result in a different fluoride adsorption mechanism on the adsorbent. Fluoride adsorption on BC was homogenous; however, on nHAST the composite was heterogeneous [36]. This result was related to the result of Medellin–Castillo and colleagues’ work. The fluoride adsorption capacity and characterization of bone char (BC) and hydroxyapatite (HAP) were compared. They reported that the point of zero charge of BC and HAP was 8.4 and 7.0, respectively. In addition, the fluoride adsorption capacity dramatically increased when the pH of the solution was decreased from 7.0 to 5.0, of which the q_BC_/q_HAP_ ratio was 0.95 and 0.77, correspondingly, at an equilibrium fluoride concentration of 1.5 mg/L [33].

The results in this study illustrate that the best charring condition for synthesizing pig bone char (PBC) and chicken bone char (CKBC) is 650 °C for 3 h, and 550 °C for 3 h for cow bone char (CBC). The CKBC produced significantly higher HAP than other BCs in this work. In contrast, CKBC had the lowest specific surface area when compared to the others. The fluoride adsorption capacity in the unit of mg fluoride per g of bone char of CBC (0.678 mg/g of BC) was higher than PBC (0.233 mg/g of BC) and CKBC (0.025 mg/g of BC), although CBC had a lower amount of hydroxyapatite than CKBC. This was due to the highest specific surface area of CBC. This indicates that the fluoride adsorption capacity was affected not only by the amount of hydroxyapatite content but also the specific surface area.

Although BCs can be applied in the defluoridation process, the problem of the yellowish color of treated water remains. In addition, BCs have a lower fluoride adsorption capacity when compared to other adsorbents, as shown in Table 5. This is related to the process for bone char generation. The calcination process was conducted in this study because it is a normal and uncomplicated process for bone char produced under a limited amount of oxygen [39]. Thus, this is the limit of BC as an adsorbent in the defluoridation process. The enhancing fluoride adsorption capacity of adsorbent will be conducted in future research.

## 5. Conclusions

Different types of BCs were produced from three types of raw bones—cow BC (CBC), chicken BC (CKBC), and pig BC (PBC)—and synthesized under different charring temperatures (450–650 °C) and durations (1–3 h). The charring temperatures and durations for PBC, CKBC, and CBC synthesis affected the HAP content in the BCs. However, the fluoride adsorption capacities of these BCs were affected not only by the HAP content but also by the specific surface area. At the optimum charring conditions (temperature and time), PBC had the highest hydroxyapatite (HAP) content (0.928 g-HAP/g-BC), while CBC had the highest specific surface area (103.11 m^2^/g-BC). CBC also had the maximum fluoride adsorption capacity (0.788 mg-F/g-HAP), suggesting that the fluoride adsorption capacity depends more on the specific surface area of the BC than the HAP content. The fluoride adsorption capacity was promoted in solutions with a pH lower than the PZC of the BC. The maximum fluoride adsorption capacity of BC reached the maximum value when the solution had a pH of approximately 6.0. The dominant mechanism of fluoride adsorption on the BC adsorbents was the exchange between the fluoride and hydroxyl ions. The fluoride ions desorbed in the alkaline solution with a pH higher than 11.0. In this study, although the CBC adsorbent exhibited the highest fluoride adsorption capacity, the treated water still had hygiene problems (a yellowish color). Thus, the PBC adsorbent is a better alternative for the defluoridation process in aqueous solutions. In the practical use of prepared material, the PBC can be applied in household filtration systems which can be set in a small column of the filtration unit to remove possible occurrences of fluoride in drinking water.

## Figures and Tables

**Figure 1 ijerph-18-06878-f001:**
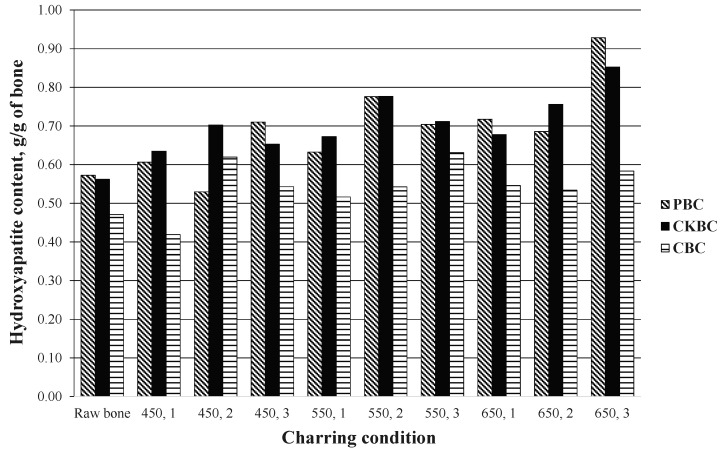
The amount of hydroxyapatite, g/g of the bone after different charring conditions.

**Figure 2 ijerph-18-06878-f002:**
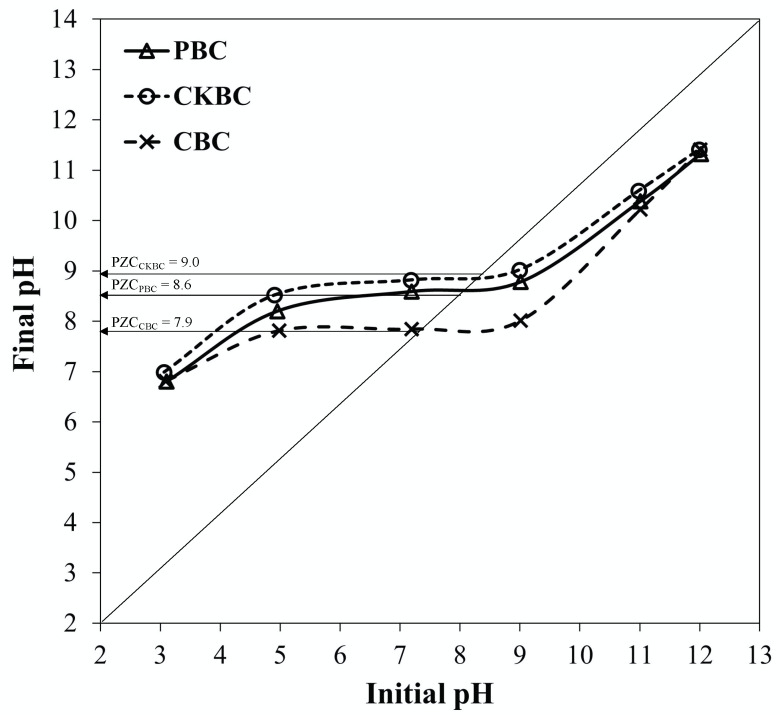
Determination of the PZC values of the PBC, CKBC, and CBC adsorbents.

**Figure 3 ijerph-18-06878-f003:**
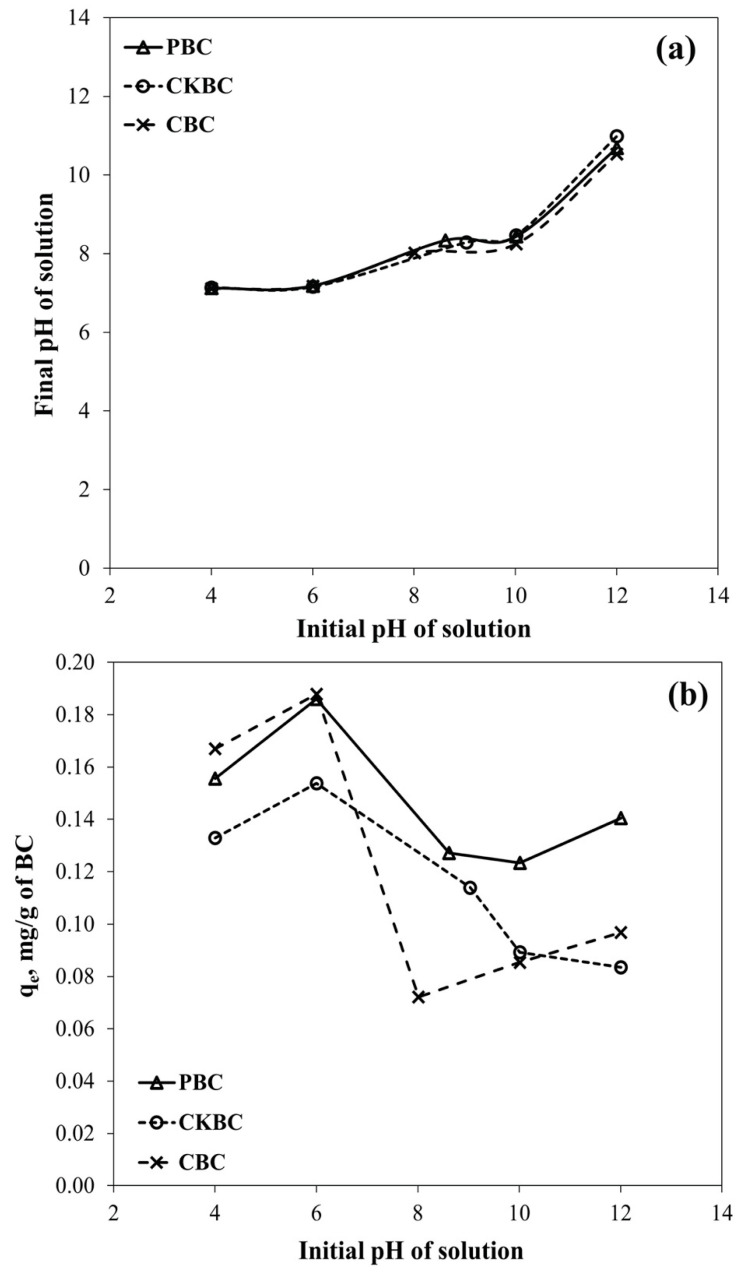
(**a**) The changing trend of the pH of the solution after fluoride adsorption process on the PBC, CKBC, and CBC adsorbents, and (**b**) the relationship between the initial pH solution and the fluoride adsorption capacity onto PBC, CKBC, and CBC.

**Figure 4 ijerph-18-06878-f004:**
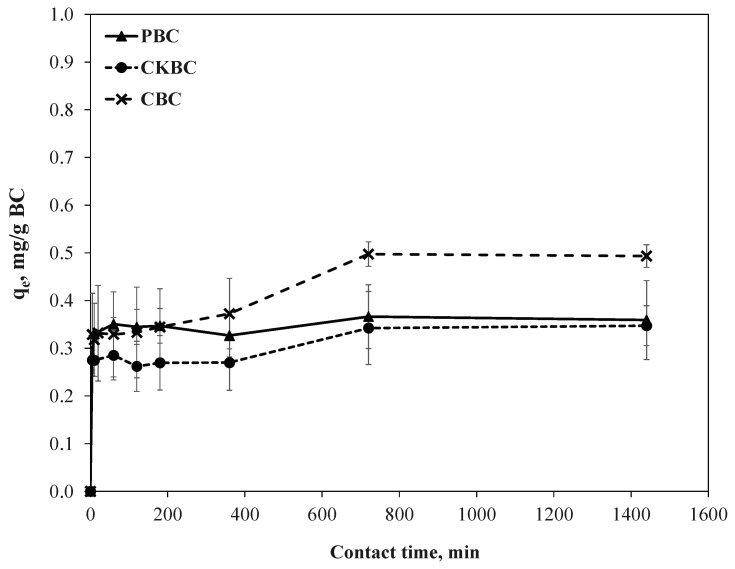
The adsorption kinetics of fluoride ions on PBC, CKBC, and CBC as a function of contact time at the initial fluoride concentration of 20 mg/L and BC dose of 20 g/L.

**Figure 5 ijerph-18-06878-f005:**
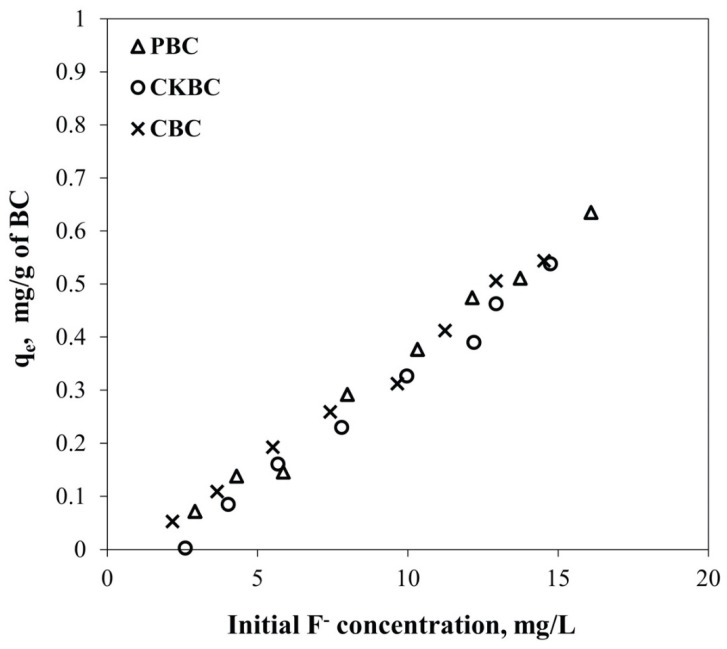
The adsorption isotherms for fluoride removal from the aqueous solution using PBC, CKBC, and CBC at a BC dose of 20 g/L with varying initial fluoride concentrations.

**Figure 6 ijerph-18-06878-f006:**
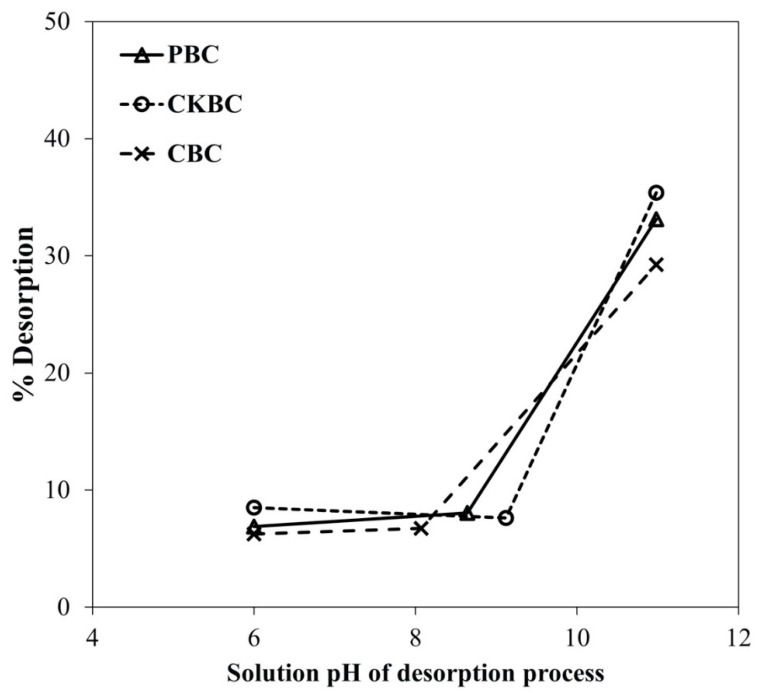
Effects of the solution pH on fluoride desorption from the PBC, CKBC, and CBC adsorbents.

**Table 1 ijerph-18-06878-t001:** Surface area, pore volume, and average pore size of BCs.

Parameters	PBC	CKBC	CBC
Specific surface area (m^2^/g)	83.79	62.80	103.11
Total pore volume (cc/g)	0.3490	0.3288	0.3353
Average pore size (Å)	83.31	104.70	65.05

**Table 2 ijerph-18-06878-t002:** Fluoride adsorption capacity of BCs.

BC Type.	HAP Content (g/g of BC)	Surface Area (m^2^/g of BC)	Specific Surface Area of HAP (m^2^/g of HAP)	q_e_ of Fluoride, (mg/g of HAP)
PBC	0.928	83.79	90.29	0.438 _(n=3)_
CKBC	0.853	62.80	73.62	0.407 _(n=3)_
CBC	0.631	103.11	163.41	0.788 _(n=3)_

**Table 3 ijerph-18-06878-t003:** Kinetic parameters of fluoride adsorption on PBC, CKBC, and CBC.

BC Type	q_e_,_exp_ (mg/g)	Pseudo-First-Order			Pseudo-Second-Order		
		q_e_,_cal_ (mg/g)	K_p1_, (min^−1^)	*R* ^2^	q_e_,_cal_ (mg/g)	K_p2_, g/(mg·min)	*R* ^2^
PBC	0.366	0.192	0.003	0.300	0.361	0.516	0.992
CKBC	0.347	0.173	0.007	0.913	0.349	0.107	0.995
CBC	0.497	1.000	0.005	0.401	0.502	0.053	0.992

**Table 4 ijerph-18-06878-t004:** Isotherm parameters of fluoride adsorption on the PBC, CKBC, and CBC.

BC Type	Langmuir		Freundlich		
	K_L_, (L/mg)	*R* ^2^	n	K_F_, (L/g)	*R* ^2^
PBC	1.99 × 10^−3^	0.792	0.532	0.051	0.640
CKBC	1.22 × 10^−3^	0.771	0.207	0.001	0.413
CBC	1.96 × 10^−3^	0.938	0.569	0.059	0.877

**Table 5 ijerph-18-06878-t005:** Comparison of fluoride adsorption capacity in different types of adsorbent.

Adsorbents	Process	q_e_, (mg/g)	References
Pig bone char	Calcination	0.366	This study
Chicken bone char	Calcination	0.347	
Cow bone char	Calcination	0.497	
Bone char	Pyrolysis	7.32	[36]
		0.85	[21]
		6.28	[38]
Hydroxyapatite	-	1.61	[14]

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
