# Peer review of "Evaluation of Fluoride Adsorption Mechanism and Capacity of Different Types of Bone Char"

_ijerph, 2021, doi:10.3390/ijerph18136878_

Round 1

Reviewer 1 Report

Manuscript entitled “Evaluation of fluoride adsorption mechanism and capacity of different types of bone char” submitted by Benyapa Sawangjang, Phacharapol Induvesa, Aunnop Wongrueng, Chayakorn Pumas, Suraphong Wattanachira, Pharkphum Rakruam, Patiparn Punyapalakul, Satoshi Takizawa and Eakalak Khan, can be accepted for publication in International Journal of Environmental Research and Public Health, after a major revision.

            Here is a list of my specific comments:

  1. General comment: The quality of figures 4-7 should be significant improved.
  2. Page 2, Introduction: “In this study, we investigated…”. At the end of Introduction, the main objectives of this study should be clear and detailed presented.
  3. Page 3, 2.3. Adsorption kinetics and isotherm of BC: “The filtrates were analyzed for residual fluoride concentration…”. The analytical method used for fluoride determination should be detailed.
  4. Page 4, 3.1.1. HAP content of BCs: “The optimum charring conditions were 650 °C charring…”. How was selected this optimal temperature?
  5. Page 7, Fig. 2: All these figures should be compressed into one.
  6. Page 9, 3.4. Effect of solution pH: This section should be moved before 3.2. Kinetic adsorption of BC adsorbents section.
  7. Page 11, 4. Discussion: This section should be detailed, and all the experimental results included in the previous section should be detailed discussed, in accordance with the main objectives of this study. In addition, a comparison of the maximum adsorption capacity of these materials and others reported in literature should be added, to highlight the importance of this study.
  8. Page 12, 5. Conclusions: This section is quite to general. Include here the most important experimental results and findings.

Reviewer 2 Report

Dear Authors,

After reading the article I have following remarks.

  1. Page 2, Materials and Methods section. The length should be given in mm, cm etc. It should be corrected. The entire manuscript body should be checked.
  2. Why did the Authors choose the given temperatures and charring time? What was the basis? It should be explained and corrected.
  3. Why the initial fluoride concentrations from 1-16 mg/L were selected? Are they typical for groundwater? It should be clearly explained.
  4. Page 9, Figure 4, x- axis. It should be F^- instead of F (fluoride instead of fluorine).
  5. In conclusions the information on practical use of prepared material should be added.

Round 2

Reviewer 1 Report

Manuscript entitled “Evaluation of fluoride adsorption mechanism and capacity of different types of bone char” submitted by Benyapa Sawangjang, Phacharapol Induvesa, Aunnop Wongrueng, Chayakorn Pumas, Suraphong Wattanachira, Pharkphum Rakruam, Patiparn Punyapalakul, Satoshi Takizawa and Eakalak Khan, can be accepted for publication in International Journal of Environmental Research and Public Health, after a major revision.

            Here is a list of my specific comments:

  1. General comment 1: The axes of Figs. 2-4 and 7 must be adjusted so that there is not so much empty space left.
  2. General comment 2: The interpretation of the experimental results is still very poor and must be improved.
  3. Page 4, Eqs. 5 and 7 should be deleted, because the linear forms are enough.
  4. Pages 7-8, Figs 3 and 4 should be included in a single figure (a and b).
  5. Page 11: “An analysis of the BC textural properties…”. This paragraph should be detailed.
  6. Page 14, Table 5: More examples should be added in this table.
  7. Page 14, Discussion: The equilibrium and kinetics data should be detailed discussed in this section.
  8. Page 14, Discussion: The experimental results obtained from desorption studies should be also discussed. 
